# Antimicrobial Susceptibility of Bacterial Pathogens from Patients with Ocular Surface Infections in Germany, 2020–2021: A Comparison with the Data from Three Previous National Studies

**DOI:** 10.3390/antibiotics13060471

**Published:** 2024-05-21

**Authors:** Esther Wohlfarth, Michael Kresken, Fabian Deuchert

**Affiliations:** Antiinfectives Intelligence GmbH, c/o Rechtsrheinisches Technologie- und Gründerzentrum, Gottfried-Hagen-Straße 60-62, 51105 Cologne, Germany; michael.kresken@antiinfectives-intelligence.de (M.K.); fabian.deuchert@antiinfectives-intelligence.de (F.D.)

**Keywords:** topical agents, susceptibility testing, ocular surface infections, ophthalmic drugs

## Abstract

Bacteria are a major cause of superficial eye infections, especially in children. The present study aimed to (i) determine the antimicrobial susceptibility patterns of ocular bacterial pathogens recovered in 20 laboratories during the period 2020–2021 and (ii) compare these results to those from three studies of the same design conducted in 2004, 2009, and 2015 in Germany. Cut-off values defined by EUCAST were used as breakpoints. A total of 1366 bacterial isolates were collected. The most frequent ocular specimens were conjunctival smears (54.3%). Susceptibility rates of *Staphylococcus aureus* (*n* = 594), *Haemophilus influenzae* (*n* = 178), and *Streptococcus pneumoniae* (*n* = 149) to chloramphenicol, gentamicin, kanamycin, neomycin, levofloxacin, ofloxacin, and oxytetracycline were >90% each. Overall, only minor changes in resistance levels were observed in the period since 2004. Therefore, all tested antimicrobials can still be recommended for local therapy of ocular surface infections.

## 1. Introduction

Acute infectious conjunctivitis is a common eye disorder in primary care. Although less common than viruses, bacteria are an important cause of conjunctivitis, particularly in children [1]. Leading bacterial pathogens of conjunctivitis are *Staphylococcus aureus*, *Streptococcus pneumoniae*, and *Haemophilus influenzae*. Compared to placebo, the use of antibiotic eye drops (e.g., fluoroquinolones or macrolides, alone or in combination with steroids) is associated with improved clinical and microbiological remission rates [2]. However, any use of antibiotics may lead to the emergence of resistant organisms [3]. It is, therefore, necessary to regularly determine the antibiotic susceptibility of bacterial pathogens in patients with ocular surface infections.

In Germany, the first multi-center study to investigate the occurrence of resistance in bacterial isolates from ocular surface infections (Ophthalmic Study I [OS1]) was carried out in 2004 [4]. A second study was performed in 2009 (Ophthalmic Study II [OS2]) and a third one (Ophthalmic Study III [OS3]) was conducted in 2015 [5,6]. There were only minor changes detected in the susceptibility rates of Gram-positive and Gram-negative bacteria against gentamicin, kanamycin, levofloxacin, ofloxacin, and chloramphenicol between 2004 and 2015. The present study (Ophthalmic Study IV [OS4]) was carried out in 2020–2021.

The objectives of the present study were to (i) determine the susceptibility of ocular isolates of *S. aureus*, *Staphylococcus epidermidis*, *S. pneumoniae*, *H. influenzae*, *Moraxella catarrhalis*, and *Pseudomonas aeruginosa*, as well as members of the order Enterobacterales, against seven antibiotics used as ophthalmic drugs, and (ii) compare the susceptibility rates to those from the previous studies (OS1–OS3). The antibiotics studied were chloramphenicol, gentamicin, kanamycin, neomycin, levofloxacin, ofloxacin, and oxytetracycline. Furthermore, the susceptibility rates of oxacillin and cefoxitin were determined to phenotype methicillin-resistant staphylococci.

## 2. Results and Discussion

Collecting isolates for this study (OS4) was heavily influenced by the COVID-19 pandemic. According to the laboratories involved in OS4, the number of eyes swabs sent for testing fell considerably after the start of the pandemic. For this reason, the collection period had to be extended several times, resulting in a total period of 21 months (1 April 2020 to 31 December 2021). The extension of the collection period was also necessary because only 20 out of 40 laboratories that were asked to participate had the capacity to take part in this study. The limitations of the study, which could potentially have an impact on the results, were missing data on comorbidities of the patients or antibiotic therapy that the patients received.

### 2.1. Bacterial Isolates and Patient Demographic Data

A total of 1366 isolates were collected. This number was slightly higher than in the three previous studies (Appendix A). More than two thirds of the isolates were obtained from outpatients and approximately 20% from inpatients, while, for the remaining isolates, the status of the respective patients was unknown or not documented. The most common sampling material for isolation was conjunctival swabs, followed by eye swabs without further specification (Figure 1; Appendix A).

The gender ratio of the patients was approximately equal. The majority of patients belonged to the ≤1 year age group, followed by 60- to 79-year-olds. Overall, just over half of the patients were younger than 10 years, with a median age of 3 years. This distribution was in accordance with previous studies. The type of infection was documented in 43.3% of cases. Of the 774 patients, approximately 84% had conjunctivitis or keratoconjunctivitis (Appendix A). The most frequently isolated pathogenic species was *S. aureus*, followed by *H. influenzae* and *S. pneumoniae* (Figure 2; Appendix A). 

### 2.2. MIC Frequency Distributions and Ratios of Susceptibility to Resistant Isolates

Cut-off values (epidemiological cut-off values (ECOFF), tentative epidemiological cut-off values ((T)ECOFF), or screening cut-off values) defined by the European Committee on Antimicrobial Susceptibility Testing (EUCAST) were used to distinguish between wild-types (hereafter designated as susceptible isolates) and non-wild-types (hereafter designated as resistant isolates) (see Section 3 and Appendix A for details). 

The minimum inhibitory concentrations (MICs) that inhibit 50% and 90% of the isolates (MIC_50_, MIC_90_), the percentage of susceptible isolates, and the percentage of resistant isolates for the antibacterial agents tested in this study, in comparison with the results of the three previous studies, are presented in Table 1, Table 2 and Table 3. MIC distributions of the antibacterial agents tested in the present study are presented in Appendix A.

#### 2.2.1. *Staphylococcus aureus*

Susceptibility results are displayed in Table 1 and Appendix A. In all, 20 out of the 594 (3.4% [95%-CI: 2.3–5.3%]) *S. aureus* isolates in the present study (OS4) had a cefoxitin MIC value > 4 mg/L, indicating a methicillin-resistant *S. aureus* (MRSA) phenotype. Of note, one MRSA isolate that was tested was cefoxitin-resistant but oxacillin-susceptible (MIC 2 mg/L). The MRSA rate was almost identical to the rate found in the 2015 study (OS3: 3.6%) but was significantly lower than the resistance rates found in 2009 (OS2: 8.4% [95%-CI: 7.4–13.0%]) and in 2005 (OS1: 9.9% [95%-CI: 7.4–13.0%]). A decline in MRSA rates was also observed in other resistance surveillance studies conducted in Germany during this period [7,8,9]. According to the European surveillance system EARS-Net, the MRSA rate in German patients with invasive infections decreased from 21.4% to 5.5% between 2005 and 2020 [7]. The MRSA rate found in the present study was also lower than the MRSA rates found for outpatients (7.0%) and inpatients (7.4%) in the 2019–2020 susceptibility study conducted by the Paul-Ehrlich-Society [8].

Resistance to fluoroquinolones decreased significantly between 2015 (OS3) and 2020–2021 (OS4, present study) (Table 1): for levofloxacin, from 11.1% (95%-CI: 8.3–14.8%) to 7.4% (95%-CI: 5.6–9.8%), and for ofloxacin, from 11.1% (95%-CI: 8.3–14.8%) to 7.6% (95%-CI: 5.7–10.0%). The resistance rate of levofloxacin corresponded to the resistance rate of outpatient isolates found in the 2019–2020 susceptibility study conducted by the Paul-Ehrlich-Society (8.1%). The isolates obtained from inpatients in the Paul-Ehrlich-Society susceptibility study showed a resistance rate of 10.7% to levofloxacin [8]. In contrast, another study from Germany and a study from the UK reported increasing fluoroquinolone resistance rates [10,11], highlighting the importance of regional resistance surveillance in ocular surface infections.

The resistance rates determined in the present study (OS4) for kanamycin (3.7%; 95%-CI: 2.5–5.5%) and gentamicin (1.9%; 95%-CI: 1.0–3.3%) remained almost unchanged compared to 2015 (OS3). However, they were significantly lower than in 2009 (OS2; kanamycin 8.1% [95%-CI: 5.8–11.2%]; gentamicin 3.8% [95%-CI: 2.3–6.2%]) and in 2004 (OS1; kanamycin 13.3% [95%-CI: 10.4–16.8%]; gentamicin 8.9% [95%-CI: 6.6–12.0%]). The gentamicin resistance rate in OS4 was slightly lower than the rate found for the isolates from inpatients in the Paul-Ehrlich-Society susceptibility study (2.7%). Gentamicin resistance in isolates obtained from outpatients was not examined in the Paul-Ehrlich-Society study [8]. Resistance to neomycin, which was not examined in studies OS1–OS3, may also have decreased, as there is almost complete cross-resistance between neomycin and kanamycin in *S. aureus*. 

Resistance to chloramphenicol decreased from 2.8% (95%-CI: 1.6–4.7%) in 2004 (OS1) to 0.3% (95%-CI: 0.1–1.2%) in 2020–2021 (OS4). Resistance to oxytetracycline remained unchanged.

As expected, some of the test compounds showed greater differences in the resistance rates between isolates of the MSSA (cefoxitin-susceptible) and MRSA phenotypes (Table 1, Appendix A). 

Between 2004 (OS1) and 2015 (OS3), the proportion of isolates with resistance to gentamicin and kanamycin in MSSA decreased from 5.9% (95%-CI: 3.9–8.6%) to 1.4% (95%-CI: 0.6–3.3%) and from 7.9% (95%-CI: 5.6–11.0%) to 2.6% (95%-CI: 1.4–4.9%), respectively. In this study (OS4), resistance rates were 1.7% for gentamicin and 2.8% for kanamycin. The level of resistance to fluoroquinolones (levofloxacin, ofloxacin) in MSSA isolates has remained approximately the same since 2004 (6–9%) (Table 1). 

Gentamicin resistance in MRSA isolates significantly decreased from 37.2% (95%-CI: 24.4–52.1%) in 2004 (OS1) to 5.0% (95%-CI: 0.9–23.6%) in this study. Kanamycin resistance also decreased significantly, from 62.8% (95%-CI: 47.9–75.6%) to 30.0% (95%–CI: 14.5–51.9%). In contrast, fluoroquinolone resistance initially increased from approximately 60% in 2004 (OS1) to over 90% in 2009 (OS2) and subsequently decreased to 35% in the present study. 

Two MSSA isolates but no MRSA isolates showed resistance to chloramphenicol. Resistance to oxytetracycline was detected in 13 MSSA and 3 MRSA isolates. Due to cross-resistance, resistance rates of neomycin were identical to those of kanamycin (Table 1 and Appendix A).

#### 2.2.2. *Staphylococcus epidermidis*

The resistance rates observed for oxacillin (32.9% [95%–CI: 23.0%–44.5%]) and gentamicin (27.1% [95%–CI: 18.1%–38.5%]) (Appendix A) in the present study were significantly lower than in 2015 (OS3: oxacillin and gentamicin each 48.5% [95%–CI: 39.0–58.1%]), but comparable to the resistance rates in 2009 (OS2: oxacillin and gentamicin each 30.8% [95%–CI: 22.2–40.9%]) (Table 2). 

EUCAST has not set a cut-off value for chloramphenicol. Applying the ECOFF for *S. aureus* (MIC > 16 mg/L), one isolate (1.4%) was classified as chloramphenicol-resistant. Resistance rates in the previous studies were 7.5% (OS1), 1.1% (OS2), and 1.0% (OS3). The level of gentamicin resistance in this study (27.1%) was comparable to that assessed in OS2 (30.8%) but lower than the level of resistance in OS1 (42.5%) and OS3 (48.5%). 

EUCAST has also not established cut-off values for kanamycin and neomycin. When comparing the MIC values, cross-resistance between kanamycin and neomycin appears to be less common in *S. epidermidis* than in *S. aureus.* The distribution of kanamycin MICs suggests a (T)ECOFF of 8 mg/L. Using this value, the proportion of resistant isolates would be 24.3% (*n* = 17). All kanamycin-resistant isolates would also be gentamicin-resistant (MIC values 4–64 mg/L), while the neomycin MIC values were ≤0.25 mg/L (*n* = 10), 0.5–2 mg/L (*n* = 5), and 16 mg/L (*n* = 2).

Resistance to fluoroquinolones was estimated based on the number of isolates with levofloxacin MICs > 0.5 mg/L (32.9%). The level of resistance to levofloxacin was approximately the same in all four studies (29–33%) (Table 2). Resistance against oxytetracycline occurred in 41.4% of the isolates (OS2: 14.3%). However, approximately half of the resistant isolates were inhibited at 4 mg/L oxytetracycline, which is right at the presumed (T)ECOFF (MIC > 2 mg/L) (Appendix A). It is also worth noting that oxytetracycline MIC values in this study were, on average, one MIC step higher than the MIC values in OS2, suggesting that the isolates inhibited at 4 mg/L oxytetracycline in the present study belong to the wild-type population.

#### 2.2.3. *Streptococcus pneumoniae*

*S. pneumoniae* usually shows comparatively low susceptibility to aminoglycosides. MIC_50/90_ values in this study were 8/8 mg/L for gentamicin, 32/64 mg/mL for kanamycin, and 64/64 mg/L for neomycin. None of the isolates in this study were chloramphenicol-resistant (OS1: 1.1%; OS2: 0.5%; OS3: 0%) (Table 2 and Appendix A) or resistant to fluoroquinolones (levofloxacin, ofloxacin; OS1: 1.6% each; OS2: 0% each; OS3: 0.4% each). Six isolates (4%) were classified as oxytetracycline-resistant (Appendix A).

Low resistance rates in *S. pneumoniae* have also been reported in other studies, for example, in an Australian study that investigated the epidemiology of microbial keratitis, and in a nationwide study in the USA (Antibiotic Resistance Monitoring in Ocular Micro-organisms—ARMOR) that monitored the development of resistance between 2009 and 2018 [12,13]. The isolates from the susceptibility study of the Paul-Ehrlich-Society 2019–2020 were also 100% susceptible to levofloxacin [8].

#### 2.2.4. *Haemophilus influenzae*

The resistance rates in *H. influenzae* remained favorable over time. Three isolates in the present study were chloramphenicol-resistant (Appendix A). The resistance rate (<2%) was comparable to those found in OS1–OS3 (Table 3). In this study, 3.9% of the isolates showed resistance to levofloxacin and ofloxacin. In comparison, resistance rates of levofloxacin and ofloxacin in the previous studies were 1.2% and 1.8% (OS1), 3.0% and 5.1% (OS2), and 3.1% and 4.0% (OS3), respectively. Gentamicin-resistant isolates were not detected, as in the previous studies. EUCAST has not yet defined cut-off values for kanamycin or neomycin. If the (T)ECOFF of kanamycin defined for *E. coli* (susceptible ≤ 16 mg/L, resistant > 16 mg/L) was applied, two isolates (1.1% [OS1: *n* = 2, 1.2%; OS2: *n* = 5, 2.1%; OS3: *n* = 3, 0.9%]) would be considered kanamycin-resistant (MICs, ≥512 mg/L). These two isolates would also be evaluated as neomycin-resistant if the neomycin ECOFF for *E. coli* (susceptible ≤ 8 mg/L, resistant > 8 mg/L) was applied. The MIC values of neomycin for the two isolates were 128 mg/L and ≥512 mg/L. Three isolates (1.7%) were oxytetracycline-resistant (MIC ≥ 128 mg/L) (OS2: *n* = 2, 0.9%), with one isolate also resistant to fluoroquinolones and another also resistant to kanamycin and neomycin.

#### 2.2.5. *Moraxella catarrhalis*

The resistance situation in *M. catarrhalis* was comparable to that in *H. influenzae*. All isolates were susceptible to chloramphenicol (OS1: 100%; OS2: 98.3%; OS3: 100%), oxytetracycline (OS2: 100%), and levofloxacin and ofloxacin (OS1–OS3: 100% each) (Appendix A and Table 3). EUCAST has not set cut-off values for gentamicin, kanamycin, or neomycin. The MIC distributions of the three aminoglycosides indicate that there are no resistant isolates (Appendix A). 

#### 2.2.6. *Pseudomonas aeruginosa*

*P. aeruginosa* is listed as an “expected resistant phenotype” for chloramphenicol, kanamycin, neomycin, and tetracyclines [14]. Two isolates (3.1%) were gentamicin-resistant (MICs ≥ 128 mg/L) (Appendix A). Both isolates were obtained from inpatients from the same geographical region. One gentamicin-resistant isolate (2.2%) was detected in OS1 but none were detected in OS2 and OS3 (Table 3). Cut-off values of levofloxacin and ofloxacin were 2 mg/L and 4 mg/L, respectively. Two isolates were classified as levofloxacin-resistant (3.1% [OS1: *n* = 6, 13.3%; OS2: *n* = 4, 11.1%; OS3: 0%), with MICs of 4 mg/L and 16 mg/L. The resistant isolate inhibited at 16 mg/L levofloxacin was also ofloxacin-resistant (MIC ≥ 32 mg/L), while the resistant isolate inhibited at 4 mg/L levofloxacin was classified as ofloxacin-susceptible because the ofloxacin MIC was also 4 mg/L (Appendix A). Ofloxacin resistance rates found in the three previous studies were consistent with those of levofloxacin (Table 3). In the United States (ARMOR study), fluoroquinolone resistance rates in *P. aeruginosa* were at a similar level, with no changes noted between 2009 and 2018 [13]. In contrast, in the susceptibility study of the Paul-Ehrlich-Society 2019–2020, more than 12% of isolates obtained from outpatients and inpatients were levofloxacin-resistant (MIC > 2 mg/L), while the proportion of isolates with a gentamicin MIC above the ECOFF of 8 mg/L was less than 2% [8]. 

#### 2.2.7. Enterobacterales

Susceptibility data of the Enterobacterales isolates are summarized in Table 3 and Appendix A. The level of resistance largely depends on the relative proportion of each Enterobacterales species. A resistance rate above 10% was only determined for ofloxacin (13.2%). Resistance to levofloxacin was detected in 8.5% of the isolates. The difference between the two resistance rates is due to the fact that the screening cut-off values for both drugs are 0.25 mg/L, but levofloxacin was, on average, twice as active as ofloxacin. MIC_50/90_ values were 0.063/0.25 mg/L (levofloxacin) and 0.125/0.5 mg/L (ofloxacin). In OS1 (*n* = 161), three isolates (1.8%) were levofloxacin-resistant and 10 (6.1%) were ofloxacin-resistant, and, in OS2 (*n* = 129), seven were levofloxacin-resistant (5.4%) and eleven were ofloxacin-resistant (8.5%). Enterobacterales were not tested in OS3.

The rates of resistance to gentamicin and neomycin in the present study were 3.4% (OS1: 1.8%; OS2: 3.1%) and 1.3%, respectively. EUCAST has not defined a cut-off value for kanamycin. Applying the *E. coli* ECOFF to kanamycin (susceptible ≤ 16 mg/L, resistant > 16 mg/L), the resistance rate would be 3.0% (OS1: 6.1%; OS2: 5.4%) (Appendix A).

Fifteen (6.4%) isolates were chloramphenicol-resistant. The resistant isolates were mainly *Proteus mirabilis* (*n* = 5) or *Serratia marcescens* (*n* = 5) (Appendix A). Resistance rates in the earlier studies were 8.5% (OS1) and 11.6% (OS2). 

EUCAST has not defined cut-off values for tetracyclines. The oxytetracycline MICs, however, showed a clear bimodal distribution (Appendix A). With a concentration of 16 mg/L as the cut-off value, 25.1% of the tested isolates would be considered resistant. However, significant differences in MIC distributions between species were observed (Appendix A). *Enterobacter cloacae* complex isolates would be 100% susceptible to oxytetracycline at 16 mg/L, while all but one of the *P. mirabilis* isolates would be oxytetracycline-resistant.

## 3. Materials and Methods

### 3.1. Study Design

The study was designed as a prospective, multi-center in vitro study, like the three previous studies. Of the 20 study centers, 10 were already involved in OS1, 17 in OS2, and 16 in OS3. The methodology of all studies remained the same, with some minor variations regarding the tested agents and species (for details see Table 1, Table 2 and Table 3). 

### 3.2. Clinical Isolates

The study sites were requested to collect 10 to 40 consecutive clinical isolates of *S. aureus*, *S. pneumoniae*, and *H. influenzae* each, from between 1 April 2020 and 31 December 2021. As for the other bacterial species (i.e., *S. epidermidis*, *M. catarrhalis*, *P. aeruginosa*) and the taxon Enterobacterales, all isolates obtained during the collection period were eligible for inclusion in the study. Only the first isolates of patients with ocular surface infections (e.g., conjunctivitis, keratitis, kerato-conjunctivitis, blepharitis, hordeolum) were permitted. The isolates were conserved until the end of the collecting period at the participating laboratories and subsequently shipped to a central laboratory (Antiinfectives Intelligence, Cologne, Germany). 

### 3.3. Confirmation of Species Identification and Susceptibility Testing

Species identification of the shipped isolates was confirmed using standard laboratory procedures, including MALDI-TOF mass spectrometry (BioMérieux GmbH, Nürtingen, Germany). Susceptibility testing was performed with nine antibacterial agents (test agents), seven of which are used as ophthalmic drugs (respective concentration ranges are given in brackets): chloramphenicol (0.125–256 mg/L), gentamicin (0.031–64 mg/L), kanamycin (0.125–256 mg/L), neomycin (0.125–256 mg/L), levofloxacin (0.008–16 mg/L), ofloxacin (0.016–16 mg/L), and oxytetracycline (0.031–64 mg/L). Additionally, staphylococci were tested for their susceptibility against cefoxitin (0.5–16 mg/L) and oxacillin (0.25–8 mg/L). MICs were determined using broth microdilution (BMD) in accordance with ISO 20776–1 (version of 2019) [15]. Industrially prepared BMD panels containing the antibacterial agents were purchased from Merlin GmbH (Bornheim, Germany). Media for MIC determination were those recommended by EUCAST [16]. Testing of non-fastidious bacteria was conducted in Cation-adjusted Mueller-Hinton-II-Broth (MHB; Becton Dickinson GmbH, Heidelberg, Germany). For *S. pneumoniae*, *H. influenzae*, and *M. catarrhalis* testing was conducted in MHB supplemented with 5% lysed horse blood (Oxoid Deutschland GmbH, Wesel, Germany) and 20 mg/L β-NAD (AppliChem GmbH, Darmstadt, Germany). The MIC values were determined using a mirror and the naked eye. Oxacillin MICs of *S. aureus* and *S. epidermidis* were determined after 24 h of incubation. 

### 3.4. Breakpoints

Due to a lack of clinical data that directly correlate therapeutic results with the MIC of pathogenic bacteria, EUCAST has not defined clinical breakpoints for topically applied antimicrobial substances. EUCAST advises one to use “regular” (clinical) breakpoints or cut-off values for the classification of bacterial phenotypes as susceptible or resistant. 

In this study, the screening cut-offs for phenotypic resistance for detection of phenotypic resistance (“screening cut-off values”) listed on page 111 of the EUCAST document “Breakpoint tables for interpretation of MICs and zone diameters Version 13.1” [17] or (tentative) epidemiological cut-off values (“[T]ECOFFs”) as shown on the EUCAST website “MIC EUCAST-EUCAST MIC and zone diameter distributions and ECOFFs” [18] were used. 

ECOFFs are set on the upper end of the MIC distribution and enable distinguishing between wild-types and non-wild-types (phenotypically resistant microorganisms). The screening cut-off values for cefoxitin and oxacillin for *S. aureus* and *S. epidermidis* can be found on page 36 of the above-mentioned EUCAST-document [18]. The data generated in the studies OS1 to OS3 were evaluated with the same breakpoints in order to be able to compare them with the results of this study.

However, we did not apply the recently published oxytetracycline ECOFF of 0.5 mg/L for *S. aureus* [18] because this value is within the recommended MIC range (0.25–1 mg/L) of the oxytetracycline-susceptible reference strain *S. aureus* ATCC 29213 (Appendix A). 

### 3.5. Quality Control

The accuracy of susceptibility testing was evaluated using reference strains: *S. aureus* ATCC 29213 (MSSA), *S. aureus* ATCC 43300 (MRSA), *S. pneumoniae* ATCC 49619, *E. coli* ATCC 25922, *P. aeruginosa* ATCC 27853, and *H. influenzae* ATCC 49766 [19]. The inoculum was quantified for every test run of the reference strains and 10% of the clinical isolates. The viable count should be approximately 2 × 10^5^ to 8 × 10^5^ CFU/mL. In all but three cases, the MIC values were within the respective quality control limits (Appendix A).

### 3.6. Statistical Evaluation

In order to evaluate statistical significance between two resistance rates R1 and R2, their 95% confidence intervals (calculated according to the Wilson method) were used [20]. Statistical significance was determined if both resistance rates lied outside of the confidence intervals of each other.

## 4. Conclusions

The time period for collecting isolates for this study (OS4) was heavily influenced by the COVID-19 pandemic. Overall, there have been only minor changes in the resistance level of the antibiotics tested since the study in 2015 (OS3). Since the first study in 2005 (OS1), the resistance situation has even improved. The level of resistance, however, largely depends on the breakpoints applied. We used ECOFF values, tentative ECOFF values, and screening cut-off values defined by EUCAST to estimate the proportion of isolates with phenotypically detectable acquired resistance mechanisms to the agents tested (non-wild-type isolates). EUCAST has not defined clinical breakpoints for the evaluation of topically applied antibiotics due to a lack of clinical data that establish a correlation between the MIC of the infecting bacteria and therapeutic results. Local application of antimicrobial agents may result in higher concentrations at the infection site as compared to systemic therapy. This also applies to the use of antibiotics to treat ocular surface infections, especially conjunctivitis. Against this background and taking into account the epidemiologically largely unchanged resistance situation, the seven tested antibiotics—chloramphenicol, gentamicin, kanamycin, neomycin, levofloxacin, ofloxacin, and oxytetracycline—can continue to be recommended for use in the local therapy of superficial eye infections. In order to analyze the pathogenicity of resistant isolates, it is planned to include therapeutic data and molecular analysis of the genetic background (e.g., virulence and resistance genes) in future studies. 

## Figures and Tables

**Figure 1 antibiotics-13-00471-f001:**
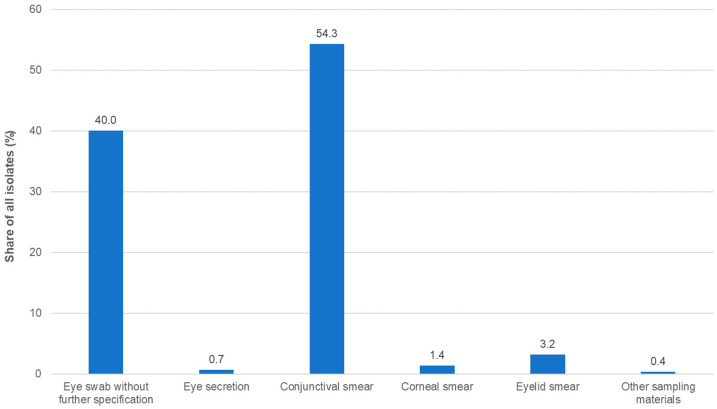
Distribution of isolates by specimen type (*n* = 1366).

**Figure 2 antibiotics-13-00471-f002:**
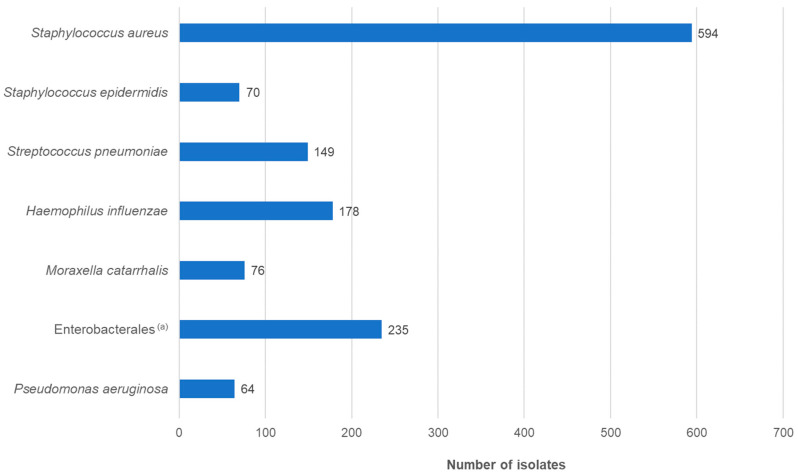
Distribution of isolates by species (*n* = 1366). ^(a)^ Enterobacterales isolates belonged to the following species: *Escherichia coli* (*n* = 50), Enterobacter cloacae complex (*n* = 32), *Klebsiella pneumoniae* (*n* = 30), *Serratia marcescens* (*n* = 27), *Proteus mirabilis* (*n* = 23), *Klebsiella oxytoca* (*n* = 22), *Citrobacter koseri* (*n* = 12), *Morganella morganii* (*n* = 11), *Klebsiella aerogenes* (*n* = 8), *Pantoea agglomerans* (*n* = 4), *Proteus vulgaris* (*n* = 3), *Raoultella ornithinolytica* (*n* = 2), *Citrobacter farmeri* (*n* = 1), *Citrobacter freundii* (*n* = 1), *Erwinia rhapontici* (*n* = 1), *Hafnia alvei* (*n* = 1), *Leclercia adecarboxylata* (*n* = 1), *Pantoea* species (*n* = 2), *Pseudoescherichia vulneris* (*n* = 1), *Raoultella planticola* (*n* = 1), *Serratia liquefaciens* (*n* = 1), *Serratia ureilytica* (*n* = 1).

**Table 1 antibiotics-13-00471-t001:** In vitro activity of the tested antimicrobial agents against *S. aureus*: comparison of the results of the Ophthalmic Study (OS4) with the results of the previous studies performed in 2004 (OS1), 2009 (OS2), and 2015 (OS3).

	Number of Isolates				Antibiotic	MIC (mg/L)								% Susceptible Isolates (Wild-Types)				% Resistant Isolates (Non-Wild-Types)			
						50%				90%											
	OS1	OS2	OS3	OS4		OS1	OS2	OS3	OS4	OS1	OS2	OS3	OS4	OS1	OS2	OS3	OS4	OS1	OS2	OS3	OS4
All	436	395	360	594	CXI ^1^	n.t.	4	4	4	n.t.	4	4	4	n.t.	91.6	96.4	96.6	n.t.	8.4	3.6	3.4
					OXA ^1^	0.5	0.5	0.5	*0.25*	2	2	1	0.5	90.1	91.6	96.4	96.8	9.9	8.4	3.6	3.2
					CHL	8	8	4	8	8	8	8	8	97.2	99.5	99.4	99.7	2.8	0.5	0.6	0.3
					GEN	0.5	0.5	0.5	0.5	2	1	0.5	1	91.1	96.2	98.1	98.1	8.9	3.8	1.9	1.9
					KAN	4	2	2	2	32	8	4	4	86.7	91.9	96.1	96.3	13.3	8.1	3.9	3.7
					NEO	n.t.	n.t.	n.t.	0.5	n.t.	n.t.	n.t.	1	n.t.	n.t.	n.t.	96.3	n.t.	n.t.	n.t.	3.7
					LEV	0.25	0.25	0.25	0.25	4	**16**	**16**	0.5	87.8	86.1	88.9	92.6	12.2	13.9	11.1	7.4
					OFL	0.5	0.5	0.25	0.5	8	**16**	16	1	86.9	86.1	88.9	92.4	13.1	13.9	11.1	7.6
					OXY	n.t.	0.25	n.t.	1	n.t.	0.5	n.t.	1	n.t.	95.7	n.t.	97.3	n.t.	4.3	n.t.	2.7
Methicillin-susceptible (MSSA) ^1^	393	362	347	574	CHL	8	8	4	8	8	8	8	8	98	100	99.4	99.7	2	0	0.6	0.3
					GEN	0.5	0.5	0.5	0.5	2	1	0.5	0.5	94.1	97.5	98.6	98.3	5.9	2.5	1.4	1.7
					KAN	4	2	2	2	8	4	4	4	92.1	96.7	97.4	97.2	7.9	3.3	2.6	2.8
					NEO	n.t.	n.t.	n.t.	0.5	n.t.	n.t.	n.t.	1	n.t.	n.t.	n.t.	97.2	n.t.	n.t.	n.t.	2.8
					LEV	0.25	0.25	0.25	0.25	0.5	0.5	0.5	0.5	93.6	93.1	91.6	93.6	6.4	6.9	8.4	6.4
					OFL	0.5	0.5	0.25	0.5	1	0.5	1	1	93.4	93.1	91.6	93.4	6.6	6.9	8.4	6.6
					OXY	n.t.	0.25	n.t.	1	n.t.	0.5	n.t.	1	n.t.	96.7	n.t.	97.7	n.t.	3.3	n.t.	2.3
Methicillin-resistant (MRSA) ^1^	43	33	13	20	CHL	8	8	8	8	16	8	8	8	90.7	93.9	100	100	9.3	6.1	0	0
					GEN	1	0.5	0.5	0.5	**64**	**32**	16	1	62.8	81.8	84.6	95	37.2	18.2	15.4	5
					KAN	**64**	**32**	2	2	**64**	**32**	**64**	**512**	37.2	39.4	61.5	70	62.8	60.6	38.5	30
					NEO	n.t.	n.t.	n.t.	0.5	n.t.	n.t.	n.t.	32	n.t.	n.t.	n.t.	70	n.t.	n.t.	n.t.	30
					LEV	16	**16**	**16**	0.5	**32**	**16**	**16**	32	34.9	9.1	15.4	65	65.1	90.9	84.6	35
					OFL	**32**	**16**	**32**	0.5	**32**	**16**	**32**	32	27.9	9.1	15.4	65	72.1	90.9	84.6	35
					OXY	n.t.	0.5	n.t.	1	n.t.	**32**	n.t.	4	n.t.	84.8	n.t.	85	n.t.	15.2	n.t.	15

The epidemiological cut-off values (ECOFF), tentative ECOFF ((T)ECOFF), or screening cut-off values were used for interpretation (see Section 3 and Appendix A for details). Abbreviations: CXI, cefoxitin; OXA, oxacillin; CHL, chloramphenicol; GEN, gentamicin; KAN, kanamycin; NEO, neomycin; LEV, levofloxacin; OFL, ofloxacin; OXY, oxytetracycline. Values presented in italics indicate the number and percentage of strains that display MICs lower or equivalent to the lowest concentration tested. Values presented in bold indicate the number and percentage of strains that display MICs higher or equivalent to the highest concentration tested. n.t., not tested; ^1^ Rates of MSSA and MRSA were determined using oxacillin as the test compound in OS1 and cefoxitin as the test compound in OS2, OS3, and OS4.

**Table 2 antibiotics-13-00471-t002:** In vitro activity of the tested substances against *S. epidermidis* and *S. pneumoniae*: comparison of the results of the Ophthalmic Study (OS4) with the results of the previous studies performed in 2004 (OS1), 2009 (OS2), and 2015 (OS3).

Species	Number of Isolates				Antibiotic	MIC (mg/L)								% Susceptible Isolates (Wild-Types)				% Resistant Isolates (Non-Wild-Types)			
						50%				90%											
	OS1	OS2	OS3	OS4		OS1	OS2	OS3	OS4	OS1	OS2	OS3	OS4	OS1	OS2	OS3	OS4	OS1	OS2	OS3	OS4
*S. epidermidis*	80	91	101	70	CXI	n.t.	*2*	*2*	2	n.t.	32	**32**	**32**	n.t.	70.6	60.4	71.4	n.t.	26.4	39.6	28.6
					OXA	2	*0.25*	*0.25*	*0.25*	**32**	**32**	**8**	**16**	48.8	69.2	51.5	68.1	51.3	30.8	48.5	32.9
					CHL	4	4	4	4	8	4	4	8	92.5	98.9	99	98.6	7.5	1.1	1	1.4
					GEN	0.125	*0.25*	*0.25*	0.125	16	**32**	**32**	32	57.5	69.2	51.5	73.6	42.5	30.8	48.5	27.1
					KAN	4	2	2	1	**64**	**32**	**64**	256	-	-	-	-	-	-	-	-
					NEO	n.t.	n.t.	n.t.	*0.125*	n.t.	n.t.	n.t.	1	n.t.	n.t.	n.t.	-	n.t.	n.t.	n.t.	-
					LEV	0.25	0.25	0.25	0.25	8	8	8	4	70	69.2	68.3	68.1	28.7	30.8	31.7	32.9
					OFL	0.5	0.5	0.5	0.5	16	**16**	16	8	-	-	-	-	-	-	-	-
					OXY	n.t.	1	n.t.	1	n.t.	16	n.t.	32	n.t.	85.7	n.t.	59.7	n.t.	14.3	n.t.	41.4
*S. pneumoniae*	187	212	240	149	CHL	2	2	2	2	4	2	2	4	98.9	99.5	100	100	1.1	0.5	0	0
					GEN	4	4	4	8	8	8	8	8	-	-	-	-	-	-	-	-
					KAN	32	16	32	32	**64**	**32**	**64**	64	-	-	-	-	-	-	-	-
					NEO	n.t.	n.t.	n.t.	64	n.t.	n.t.	n.t.	64	n.t.	n.t.	n.t.	-	n.t.	n.t.	n.t.	-
					LEV	1	0.5	0.5	1	1	1	1	1	98.4	100	99.6	100	1.6	0	0.4	0
					OFL	2	1	1	2	2	2	2	2	98.4	100	99.6	100	1.6	0	0.4	0
					OXY	n.t.	0.25	n.t.	0.25	n.t.	0.5	n.t.	0.5	n.t.	92.5	n.t.	96.0	n.t.	7.5	n.t.	4.0

The epidemiological cut-off values (ECOFF), tentative ECOFF ((T)ECOFF), or screening cut-off values were used for interpretation (see Section 3 and Appendix A for details). Abbreviations: CXI, cefoxitin; OXA, oxacillin; CHL, chloramphenicol; GEN, gentamicin; KAN, kanamycin; NEO, neomycin; LEV, levofloxacin; OFL, ofloxacin; OXY, oxytetracycline. Values presented in italics indicate the number and percentage of strains that display MICs lower or equivalent to the lowest concentration tested. Values presented in bold indicate the number and percentage of strains that display MICs higher or equivalent to the highest concentration tested. n.t., not tested; -, no cut-off value defined.

**Table 3 antibiotics-13-00471-t003:** In vitro activity of the tested substances against Gram-negative bacteria: comparison of the results of the Ophthalmic Study (OS4) with the results of the previous studies performed in 2004 (OS1), 2009 (OS2), and 2015 (OS3).

Species	Number of Isolates				Antibiotic	MIC (mg/L)								% Susceptible Isolates (Wild-Types)				% Resistant Isolates (Non-Wild-Types)			
						50%				90%											
	OS1	OS2	OS3	OS4		OS1	OS2	OS3	OS4	OS1	OS2	OS3	OS4	OS1	OS2	OS3	OS4	OS1	OS2	OS3	OS4
*H. influenzae*	164	234	325	178	CHL	0.5	*0.25*	*0.5*	0.5	0.5	0.5	*0.5*	1	100	98.3	99.1	98.2	0	1.7	0.9	1.7
					GEN	2	0.5	1	0.5	2	1	1	1	100	100	100	100	0	0	0	0
					KAN	2	1	2	2	4	2	4	4	-	-	-	-	-	-	-	-
					NEO	n.t.	n.t.	n.t.	2	n.t.	n.t.	n.t.	2	n.t.	n.t.	n.t.	-	n.t.	n.t.	n.t.	-
					LEV	*0.016*	*0.031*	*0.063*	0.016	0.031	*0.031*	*0.063*	0.031	98.8	97	96.9	96.1	1.2	3	3.1	3.9
					OFL	0.031	*0.063*	*0.063*	0.031	0.063	*0.063*	*0.063*	0.063	98.2	94.9	96	96.1	1.8	5.1	4	3.9
					OXY	n.t.	*0.125*	n.t.	0.5	n.t.	0.25	n.t.	0.5	n.t.	99.6	n.t.	98.3	n.t.	0.9	n.t.	1.7
*M. catarrhalis*	33	59	50	76	CHL	0.5	0.5	*0.5*	0.5	1	0.5	1	0.5	100	98.3	100	100	0	1.7	0	0
					GEN	0.25	0.5	*0.25*	0.125	0.5	0.5	*0.25*	0.5	-	-	-	-	-	-	-	-
					KAN	1	1	1	0.5	2	2	1	1	-	-	-	-	-	-	-	-
					NEO	n.t.	n.t.	n.t.	0.25	n.t.	n.t.	n.t.	0.25	n.t.	n.t.	n.t.	-	n.t.	n.t.	n.t.	-
					LEV	0.063	0.063	*0.063*	0.063	0.063	0.063	*0.063*	0.063	100	100	100	100	0	0	0	0
					OFL	0.063	0.125	0.125	0.125	0.125	0.125	0.125	0.125	100	100	100	100	0	0	0	0
					OXY	n.t.	0.25	n.t.	0.5	n.t.	0.5	n.t.	0.5	n.t.	100	n.t.	100	n.t.	0	n.t.	0
*P. aeruginosa*	45	36	34	64	CHL	64	**64**	**64**	64	128	**64**	**64**	256	-	-	-	-	-	-	-	-
					GEN	2	2	1	2	4	4	2	2	97.8	100	100	96.9	2.2	0	0	3.1
					KAN	**64**	**32**	**64**	64	**64**	**32**	**64**	128	n.a.	n.a.	n.a.	98.4	n.a.	n.a.	n.a.	1.6
					NEO	n.t.	n.t.	n.t.	8	n.t.	n.t.	n.t.	16	n.t.	n.t.	n.t.	100	n.t.	n.t.	n.t.	0
					LEV	0.5	0.5	0.5	0.5	4	4	1	1	86.7	88.9	100	96.9	13.3	11.1	0	3.1
					OFL	1	1	1	1	8	8	2	2	86.7	88.9	100	98.4	13.3	11.1	0	1.6
					OXY	n.t.	16	n.t.	16	n.t.	**32**	n.t.	32	n.t.	n.a.	n.t.	100	n.t.	n.a.	n.t.	0
Enterobacterales	164	129	n.t.	235	CHL	8	4	n.t.	8	16	32	n.t.	16	91.5	88.4	n.t.	93.6	8.5	11.6	n.t.	6.4
					GEN	0.5	0.5		0.5	1	1		1	98.2	96.9		96.6	1.8	3.1		3.4
					KAN	2	2		2	8	4		4	-	-		-	-	-		-
					NEO	n.t.	n.t.		0.5	n.t.	n.t.		2	n.t.	n.t.		98.7	n.t.	n.t.		1.3
					LEV	0.063	0.063		0.063	0.125	0.125		0.25	98.2	94.6		91.5	1.8	5.4		8.5
					OFL	0.125	0.125		0.125	0.25	0.25		0.5	93.9	91.5		86.8	6.1	8.5		13.2
					OXY	n.t.	2		4	n.t.	**32**		128	n.t.	-		-	n.t.	-		-

The epidemiological cut-off values (ECOFF), tentative epidemiological cut-off values ((T)ECOFF), or screening cut-off values were used for interpretation (see Section 3 and Appendix A for details). Abbreviations: CHL, Chloramphenicol; GEN, Gentamicin; KAN, Kanamycin; NEO, Neomycin; LEV, Levofloxacin; OFL, Ofloxacin; OXY, Oxytetracycline. Values presented in italics indicate the number and percentage of strains that display MICs lower or equivalent to the lowest concentration tested. Values presented in bold indicate the number and percentage of strains that display MICs higher or equivalent to the highest concentration tested. n.t., not tested; -, no cut-off value defined.

## Data Availability

Data are contained within the article and Appendix A.

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
