# Peer review of "Antimicrobial Susceptibility of Bacterial Pathogens from Patients with Ocular Surface Infections in Germany, 2020–2021: A Comparison with the Data from Three Previous National Studies"

_antibiotics, 2024, doi:10.3390/antibiotics13060471_

Round 1

Reviewer 1 Report

Comments and Suggestions for Authors

This work focused on the analysis of susceptibility to different antibiotics of species of microorganisms that cause ocular pathologies. The work alludes to previous analyzes focused on the same objective. Its sample size is adequate and therefore it is pertinent to expect robust statistical inferences.

Additionally, there are some considerations in this regard:

1. The collection of samples for a longer period due to the lack of swabs during the Pandemic period, in my opinion, is not a cause that could be a cause of bias in the data analysis due to the age of the data with those that are compared date back to 2004. In this sense, the number of mutations that could have accumulated in the selected microorganisms and that could contain resistance mechanisms are negligible.

2. The differences found in the quantification of resistance to different antibiotics may be caused by sampling. In this sense, there are variants in the genetic context of the bacteria that can affect the attenuation process in the presence of the antibiotic. It can also be affected by the phenomena that promote the transport of resistance genes.

3. In this sense, one of the analyzes that could complement these works (considering the sample sizes and the clinical records of the symptoms they cause) refers to the study of molecular markers in the isolates that provide information on the variability between the different resistant isolates for each of the selected species. This would help determine differences in genomic DNA (for example, directing the analysis to previously defined pathogenicity or virulence genes or resistance genes, which could be done through a simple PCR analysis). The analysis of these data would open a window of opportunity to establish comparative criteria between molecular markers and resistance and/or pathogenicity. This data analysis provides an interesting perspective in reference to the potential horizontal genetic transfer that could lead to a substantial increase in resistance and virulence. I understand that some of these ideas should be incorporated into the conclusion in order to be considered for future studies.

Cheung GYC, Bae JS, Otto M. Pathogenicity and virulence of Staphylococcus aureus. Virulence. 2021 Dec;12(1):547-569. doi: 10.1080/21505594.2021.1878688. PMID: 33522395; PMCID: PMC7872022.

Rasmi, A.H., Ahmed, E.F., Darwish, A.M.A. et al. Virulence genes distributed among Staphylococcus aureus causing wound infections and their correlation to antibiotic resistance. BMC Infect Dis 22, 652 (2022). https://doi.org/10.1186/s12879-022-07624-8

Chang CC, Gilsdorf JR, DiRita VJ, Marrs CF. Identification and genetic characterization of Haemophilus influenzae genetic island 1. Infect Immun. 2000 May;68(5):2630-7. doi: 10.1128/IAI.68.5.2630-2637.2000. PMID: 10768954; PMCID: PMC97469.

Bae S, Lee J, Lee J, Kim E, Lee S, Yu J, Kang Y. Antimicrobial resistance in Haemophilus influenzae respiratory tract isolates in Korea: results of a nationwide acute respiratory infections surveillance. Antimicrob Agents Chemother. 2010 Jan;54(1):65-71. doi: 10.1128/AAC.00966-09. Epub 2009 Nov 2. PMID: 19884366; PMCID: PMC2798543.

4. It is necessary to incorporate therapeutic data in future studies since they are elements to determine pathogenicity with virulence and antimicrobial resistance.

5. An element that is relevant in this type of analysis refers to the fact that the formulation used to add the antibiotic in each clinical case is made in a liquid medium on the ocular surface. This context makes it easier for these microorganisms to be in conditions of special susceptibility to antibiotics since the concentration is high, and the possibility of very dense biofilms forming is low. This scenario contributes to the area/volume ratio of the bacterial aggregates being very large, facilitating contact with the antibiotic. These conditions make it very difficult for the antibiotic to reach a sub-lethality condition and therefore facilitate the adaptation processes. In this sense, it is likely that resistant isolates have very efficient enzymatic mechanisms to quickly cancel or attenuate the effect of the antimicrobial. (which could be detected by PCR)

Author Response

Thank you very much for your detailed review and the interesting background information.

We included another sentence in the manuscript to highlight the fact that molecular analysis will be necessary in future studies to get more information about the distribution of virulence genes and resistance genes in these isolates.

Reviewer 2 Report

Comments and Suggestions for Authors

Dear author

It is a well-written and presented manuscript.

I wish you all the best.

Regards,

Author Response

Thank you very much!

Reviewer 3 Report

Comments and Suggestions for Authors

The manuscript by Wohlfarth et al and colleagues investigated the antimicrobial susceptibility patterns of ocular bacterial pathogens during the period 2020-2021 and compared them to three different studies conducted in 2004,2009 and 2015 in Germany.

I have some minor concerns:

1.       Author did not mention anything about the inclusion and exclusion criteria for patient recruitment.

2.       The clinical background of the patients can influence the study especially current or past Covid exposure, diabetic patients, or any other underlying conditions.

3.       The criteria should consider the antibiotic intake of patients like no antibiotic for past week or 10 days.

4.       Table S1 should mention more parameters.

5.       Does the methods consider for this study are like earlier studies. Author should mention all the minor or major variations in methodology that can influence the results between all studies.

6.       It would be great if author can discuss in conclusion about the current recommended range of prescribed antibiotics in the clinics and corelating with the current MICs results to know the clinical relevance and outcome of study.

Author Response

Thank you very much for your detailed review. Please find our answers below your comments:

  1. Author did not mention anything about the inclusion and exclusion criteria for patient recruitment.

There were no specific inclusion/exclusion criteria determined.

  1. The clinical background of the patients can influence the study especially current or past Covid exposure, diabetic patients, or any other underlying conditions.

Information on comorbidities were not documented. We included this in the manuscript (line 57).

  1. The criteria should consider the antibiotic intake of patients like no antibiotic for past week or 10 days.

Information on antibiotic therapy was not documented. We included this in the manuscript (line 57).

  1. Table S1 should mention more parameters.

Which parameters? Please specify.

  1. Does the methods consider for this study are like earlier studies. Author should mention all the minor or major variations in methodology that can influence the results between all studies.

The methods were the same, with some minor variations in the tested antibiotics. We included this in the manuscript (line 260).

  1. It would be great if author can discuss in conclusion about the current recommended range of prescribed antibiotics in the clinics and corelating with the current MICs results to know the clinical relevance and outcome of study.

As we are talking about topical agents official recommendations are rare (at least in Germany). We could ask the involved companies if there are official guidelines available of which we were not aware of so far, and if these have changed over time. We would need more time to include this in a revised manuscript.

Reviewer 4 Report

Comments and Suggestions for Authors

The authors have conducted a followup study to evaluate the resistance patterns of bacteria in ocular surface infections in Germany. They have found that the overall resistance has decreased for specific antibiotics, and recommend that the current line of treatments should remain effective based on the data. As the authors note, the study was impacted by the lack of clinical samples during the pandemic. As such, it was extended to collect enough samples. 

The authors note that the majority of the patient groups were infants (<1 year) and the elderly. Is this true also for the previous studies? Did the pandemic have any impact on the age group distributions?  

Author Response

Thank you very much for these interesting questions. The same distribution of patient groups was observed during the last three studies (see Table S1). The distribution of patient groups remained stable during the pandemic. We have included another sentence in the manuscript (line 70: This distribution was in accordance with the previous studies.).

Reviewer 5 Report

Comments and Suggestions for Authors

Reviewer’s comments:

The authors have mentioned details about the ocular surface eye infection causing bacteria and there susceptibility patterns against the antibiotics. The authors have also compared their study to the previously done study.  

1.      Line 27 & 28: What is being compared to Placebo? Mention about its details

2.      Recent studies are not taken into consideration in the introduction section.

3.      Graphs inserted in Results section can be improved by using other type of charts such as Pie chart.

4.      Materials & methods should be discussed before results to give more clarity to reader.

5.      There are many grammatical errors in the manuscript.

6.      Observations and results are not clearly summarized in conclusion.

7.      Future prospects from the current study are missing.  

8. Plag check is recommended 

Comments on the Quality of English Language

Moderate editing of English language required

Author Response

Thank you very much for your review.

Please find our comments below:

  1. Line 27 & 28: What is being compared to Placebo? Mention about its details

We included more details in the sentence (line 29).

  1. Recent studies are not taken into consideration in the introduction section.

We included another sentence in the introduction to give an overview about the results of the first three studies (line 37).

  1. Graphs inserted in Results section can be improved by using other type of charts such as Pie chart.

We tried pie charts before and it made it even worse. Therefore, we would like to stick with the type of graphs we included in the manuscript.

  1. Materials & methods should be discussed before results to give more clarity to reader.

The format that we used is the format provided by the journal.

  1. There are many grammatical errors in the manuscript.

From our point of view there are no grammatical errors. However, as we are not native speakers, we might be wrong. If you could locate those errors we can correct them.

  1. Observations and results are not clearly summarized in conclusion.

We do not see the difference between “observations” and “results”. If you could specify your comment it would help us a lot. Thank you.